# Prevalence of SARS-CoV-2 antibodies in France: results from nationwide serological surveillance

Stéphane Le Vu [1,16 ✉], Gabrielle Jones [1,16], François Anna[2,16], Thierry Rose[3,16], Jean-Baptiste Richard [4], Sibylle Bernard-Stoecklin[1], Sophie Goyard[3], Caroline Demeret [5], Olivier Helynck[6], Nicolas Escriou[7], Marion Gransagne[7], Stéphane Petres[8], Corinne Robin[9], Virgile Monnet[10], Louise Perrin de Facci [11], Marie-Noelle Ungeheuer[11], Lucie Léon[12], Yvonnick Guillois[13], Laurent Filleul[14], Pierre Charneau[2], Daniel Lévy-Bruhl[1], Sylvie van der Werf [5,15,17] & Harold Noel [1,17]

Assessment of the cumulative incidence of SARS-CoV-2 infections is critical for monitoring the course and extent of the COVID-19 epidemic. Here, we report estimated seroprevalence in the French population and the proportion of infected individuals who developed neutralising antibodies at three points throughout the first epidemic wave. Testing 11,000 residual specimens for anti-SARS-CoV-2 IgG and neutralising antibodies, we find nationwide seroprevalence of 0.41% (95% CI: 0.05–0.88) mid-March, 4.14% (95% CI: 3.31–4.99) mid-April and 4.93% (95% CI: 4.02–5.89) mid-May 2020. Approximately 70% of seropositive individuals have detectable neutralising antibodies. Infection fatality rate is 0.84% (95% CI: 0.70–1.03) and increases exponentially with age. These results confirm that the nationwide lockdown substantially curbed transmission and that the vast majority of the French population remained susceptible to SARS-CoV-2 in May 2020. Our study shows the progression of the first epidemic wave and provides a framework to inform the ongoing public health response as viral transmission continues globally.

[1] Infectious Diseases Division, Santé publique France, Saint-Maurice, France. [2] Unit of Molecular Virology and Vaccinology, Virology Department, Theravectys, Institut Pasteur, Paris, France. [3] Unit of Lymphocyte Cell Biology, Immunology Department, INSERM 1221, Institut Pasteur, Paris, France. [4] Data Sciences Division, Santé publique France, Saint-Maurice, France. [5] Unit of Molecular Genetics of RNA Viruses, UMR 3569 CNRS, University of Paris-Diderot, Institut Pasteur, Paris, France. [6] Unit of Chemistry and Biocatalysis, UMR 3523 CNRS, Institut Pasteur, Paris, France. [7] Innovation Laboratory: Vaccines, Institut Pasteur, Paris, France. [8] Production and Purification of Recombinant Proteins Technological Platform, Institut Pasteur, Paris, France. [9] Cerba Healthcare Division, Cerba Xpert, St Ouen L'Aumone, France. [10] Eurofins Biomnis Sample Library, Eurofins Biomnis, Lyon, France. [11] ICAReB Biobanking Platform, Center for Translational Science, Institut Pasteur, Paris, France. [12] Regional Office—French Caribbean, Santé publique France, Gourbeyre, France. [13] Regional Office—Brittany, Santé publique France, Rennes, France. [14] Regional Office—Nouvelle Aquitaine, Santé publique France, Bordeaux, France. [15] National Reference Center for Respiratory Infections Viruses Including Influenza, Institut Pasteur, Paris, France. [16] These authors contributed equally: Stéphane Le Vu, Gabrielle Jones, François Anna, Thierry Rose. [17] These authors jointly supervised this work: Sylvie van der Werf, Harold Noel. ✉email: stephane.levu@gmail.com

After the first case of SARS-CoV-2 infection was reported in France on 24 January 2020, authorities largely relied on confirmed case counts to monitor the unfolding epidemic[1]. Case-based surveillance focused primarily on symptomatic patients or those with severe disease and access to biological confirmation was initially limited. The surge in COVID-19 hospitalisations and deaths, particularly in the eastern and Paris regions, led the French authorities to implement a national lockdown from 17 March to 11 May 2020.

It is now clear that a substantial fraction of infected individuals develop mild symptoms or even remain asymptomatic[2–5]. For this reason, the actual proportion of the French population infected during the first epidemic wave remains elusive. Prevalence of previous or current infections is critical to understanding the course and extent of the epidemic.

Since a serological response is likely to take place in all SARS-CoV-2 infected individuals, the corresponding serological markers should persist for at least some time. Accordingly, prevalence of SARS-CoV-2 antibodies can assess cumulative population incidence. Such an assessment can be obtained from seroepidemiological studies, provided that the antibody detection method is accurate enough, even in a low prevalence context, and that the results from the study sample can reasonably be extrapolated to the population. In addition, such studies can measure the proportion of infected individuals who developed neutralising and potentially protective antibodies, which is particularly important in the absence of a vaccine[6]. To the best of our knowledge, few seroprevalence studies have included detection of SARS-CoV-2 neutralising antibodies, and none at a national level[2,7–10].

To estimate the fraction of the French population infected with SARS-CoV-2 over time as well as the proportion of individuals having developed neutralising antibodies, we implemented serological surveillance based on serial cross-sectional sampling of residual sera obtained from clinical laboratories. Here, we present nationwide estimates of seroprevalence in the French population, with estimates stratified by age, sex and region, from three collection periods prior to, during, and following the lockdown.

## Results

**Sampled population.** A total of 9184 residual sera for Metropolitan France were randomly selected from available sera at the three collection periods (3221 samples from 9 to 15 March 2020, 3084 samples from 6 to 12 April 2020 and 2879 samples from 11 to 17 May 2020). For the French overseas departments, 613, 511 and 713 samples were included, respectively, for the three collection periods (we excluded Mayotte Island from the analysis due to an insufficient number of available samples). The age, sex and regional distribution of the sample population is shown in Supplementary Table 2.

**National seroprevalence estimates.** Nationwide seroprevalence of SARS-CoV-2 infections increased from 0.41% (95% CI: 0.05–0.88) to 4.14% (95% CI: 3.31–4.99) and 4.93% (95% CI: 4.02–5.89) between 15 March, 12 April and 17 May 2020, corresponding to 3,292,000 (95% CI: 2,685,000–3,934,000) people having been infected as of 17 May (Supplementary Table 4). When taking into account the inherent delay between infection and IgG-mediated antibody responses, this estimate provides the number of infections which occurred ~2 weeks prior to the collection periods[11]. The prevalence of pseudo-neutralising antibodies for SARS-CoV-2 S-protein rose from 0.06% (95% CI: 0.00–0.17) to 3.33% (95% CI: 2.66–4.07) over the same period (Table 1). The raw proportions of positive sera for each individual test are detailed in Supplementary Table 3. Seroprevalence increased significantly between March and April, with a ten-fold increase in relative risk, but plateaued from April to May 2020 (Fig. 1a).

**Seroprevalence by sex, age and region.** Risk did not differ by sex and seroprevalence was estimated at 5.37% (95% CI: 4.27–6.55) for men and 4.51% (95% CI: 3.57–5.54) for women at 17 May 2020 (Fig. 1a). At the same time point, 3.70% (95% CI: 2.87–4.65) of male and 2.98% (95% CI: 2.26–3.81) of female individuals had detectable pseudo-neutralising antibodies.

From mid-March to mid-May 2020, the prevalence of infections increased markedly in all age groups (Fig. 1b). As of 17 May, 1 week after the end of lockdown, the prevalence was highest among the 50–59 and 60–69 years olds (6.06%, 95% CI: 4.43–8.04 and 6.04%, 95% CI: 4.40–8.06, respectively), and lowest in children under 10 years of age (2.72%, 95% CI: 1.10–4.87). Prevalence of pseudo-neutralising antibodies followed similar trends and varied according to age from 1.59% (95% CI: 0.52–3.13) in children under 10 years old to 4.92% (95% CI: 3.36–6.89) in 40–49 year olds, then decreasing in older age groups (Table 1). Regional seroprevalence was highest in Île-de-France which includes Paris (8.82%, 95% CI: 6.90–11.01) and Grand-Est (8.56%, 95% CI: 5.83–11.82) in the week after lockdown was lifted and varied from 2.40 to 4.44% in the remaining Metropolitan regions with a clear East-West gradient (Fig. 1c). Seroprevalence in the four overseas regions ranged from 2.40% (95% CI: 1.18–3.93) in Martinique to 7.14% (95% CI: 3.96–11.50) in French Guiana (Fig. 1c). Regional prevalence estimates of pseudo-neutralising antibodies followed similar trends and were highest in Île-de-France (7.25%, 95% CI: 5.51–9.36) and in Grand-Est (7.03%, 95% CI: 4.48–10.06). Estimates ranged from 1.20 to 3.03% for the remaining Metropolitan regions and from 0.86 to 2.66% in overseas regions (Supplementary Table 4).

**Underreporting, fatality and hospitalisation rates.** We infer that as of 17 May 2020, 1 in 24 (95% CI: 19–28) cumulative infections was reported as a confirmed case. Overall infection fatality rate (IFR) for SARS-CoV-2 was estimated at 0.84% (95% CI: 0.70–1.03) and at 0.54% (95% CI: 0.45–0.66) when excluding deaths occurring in nursing homes. Infection hospitalisation rate (IHR) was estimated at 2.58% (95% CI: 2.16–3.17). Both IFR and IHR estimates show a slightly higher risk in younger children, then increase exponentially with age, peaking in ≥80 years old at 9.70% (95% CI: 7.24–13.64) and 13.96% (95% CI: 10.42–19.62), respectively (Fig. 2c).

## Discussion

Nationwide serological surveillance in France measures the extent of the epidemic during a period when case-based surveillance prioritised assessment of symptomatic cases and testing capacity was limited. We show that following the first wave of the COVID-19 epidemic, seroprevalence remained low, with about 5% of the population having developed a detectable humoral response to the virus. This level is within the same order of magnitude as studies carried out at comparable epidemic stages in Europe[4,12,13]. Estimates at multiple points of the French epidemic show a sharp increase between the first two collection periods, immediately preceding and during the generalised lockdown, followed by little progression observed at the final collection period just after lockdown ended. This confirms its substantial impact in almost halting community transmission.

The overall IFR estimated from hospitalised deaths is in line with previous estimates[14,15], but is greatly increased when accounting for deaths in nursing homes, as is found in other countries[16]. Biological analyses of the institutionalised elderly population in France are typically carried out in clinical laboratories and as such this population should be represented in our sampling and seroprevalence estimates. As IFR is not solely determined by the pathogenesis and may evolve as health systems

**Table 1 Estimated prevalence of SARS-CoV-2 neutralising antibodies in the French population from March-May 2020.**

| | 9–15 March 2020 | | 6–12 April 2020 | | 11–17 May 2020 | |
|---|---|---|---|---|---|---|
| | P (%) | 95% CI | P (%) | 95% CI | P (%) | 95% CI |
| Overall | 0.1 | 0.0; 0.2 | 2.6 | 2.1; 3.2 | 3.3 | 2.7; 4.1 |
| Sex | | | | | | |
| Male | 0.1 | 0.0; 0.2 | 2.9 | 2.2; 3.7 | 3.7 | 2.9; 4.7 |
| Female | 0.1 | 0.0; 0.2 | 2.4 | 1.8; 3.0 | 3.0 | 2.3; 3.8 |
| Age group, years | | | | | | |
| 0–9 | 0.0 | 0.0; 0.1 | 1.3 | 0.4; 2.4 | 1.6 | 0.5; 3.1 |
| 10–19 | 0.1 | 0.0; 0.2 | 2.6 | 1.6; 3.9 | 3.3 | 2.1; 4.8 |
| 20–29 | 0.1 | 0.0; 0.2 | 2.5 | 1.5; 3.7 | 3.2 | 1.9; 4.8 |
| 30–39 | 0.0 | 0.0; 0.1 | 1.8 | 1.0; 2.8 | 2.3 | 1.3; 3.6 |
| 40–49 | 0.1 | 0.0; 0.3 | 3.9 | 2.6; 5.4 | 4.9 | 3.4; 6.9 |
| 50–59 | 0.1 | 0.0; 0.3 | 3.6 | 2.4; 4.9 | 4.5 | 3.1; 6.2 |
| 60–69 | 0.1 | 0.0; 0.2 | 2.8 | 1.9; 4.0 | 3.6 | 2.4; 5.1 |
| 70–79 | 0.1 | 0.0; 0.2 | 2.3 | 1.5; 3.4 | 3.0 | 1.9; 4.2 |
| ≥80 | 0.1 | 0.0; 0.2 | 2.5 | 1.6; 3.7 | 3.2 | 2.1; 4.5 |
| Regions | | | | | | |
| Guadeloupe | 0.0 | 0.0; 0.1 | 1.3 | 0.5; 2.4 | 1.7 | 0.7; 3.1 |
| Martinique | 0.0 | 0.0; 0.1 | 0.7 | 0.2; 1.4 | 0.9 | 0.3; 1.8 |
| French Guiana | 0.1 | 0.0; 0.2 | 2.1 | 0.8; 4.3 | 2.7 | 1.0; 5.4 |
| La Reunion | 0.0 | 0.0; 0.1 | 0.9 | 0.2; 2.2 | 1.2 | 0.3; 2.8 |
| Île-de-france | 0.1 | 0.0; 0.4 | 5.7 | 4.3; 7.4 | 7.3 | 5.5; 9.4 |
| Centre-Val-de-Loire | 0.0 | 0.0; 0.1 | 1.0 | 0.3; 2.3 | 1.3 | 0.4; 2.8 |
| Bourgogne-Franche Comté | 0.1 | 0.0; 0.2 | 2.4 | 1.1; 4.2 | 3.0 | 1.4; 5.4 |
| Normandie | 0.0 | 0.0; 0.1 | 1.5 | 0.7; 2.8 | 1.9 | 0.9; 3.5 |
| Hauts-de-France | 0.0 | 0.0; 0.1 | 1.9 | 1.0; 3.1 | 2.4 | 1.3; 3.9 |
| Grand-Est | 0.1 | 0.0; 0.4 | 5.6 | 3.5; 8.1 | 7.0 | 4.5; 10.1 |
| Pays de la Loire | 0.0 | 0.0; 0.1 | 1.7 | 0.7; 3.2 | 2.2 | 0.9; 4.1 |
| Bretagne | 0.0 | 0.0; 0.1 | 1.1 | 0.3; 2.4 | 1.4 | 0.4; 3.1 |
| Nouvelle-Aquitaine | 0.0 | 0.0; 0.1 | 1.1 | 0.5; 2.0 | 1.4 | 0.7; 2.5 |
| Occitanie | 0.0 | 0.0; 0.1 | 0.9 | 0.3; 1.8 | 1.2 | 0.4; 2.4 |
| Auvergne-Rhône-Alpes | 0.1 | 0.0; 0.2 | 2.2 | 1.2; 3.6 | 2.8 | 1.5; 4.5 |
| Provence-Alpes-Côte d'Azur | 0.0 | 0.0; 0.1 | 1.2 | 0.6; 2.0 | 1.5 | 0.7; 2.6 |
| Corse | 0.0 | 0.0; 0.1 | 1.1 | 0.3; 2.7 | 1.4 | 0.4; 3.5 |

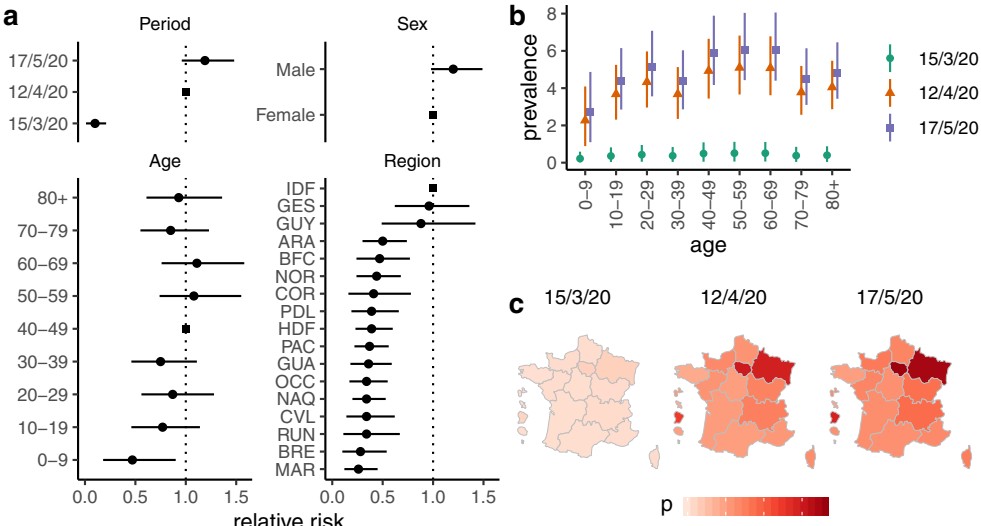

**Fig. 1 Prevalence of SARS-CoV-2 antibodies in France. a** Estimated relative risks of seroprevalence by collection period, sex, age and region. Dots represent mean relative risk and bars 95% uncertainty interval of relative risk estimate over $10^4$ iterations. Reference categories are indicated by a square. Collection periods are indicated by last day of each week (9–15 March 2020, 6–12 April 2020 and 11–17 May 2020). Regions: GUA Guadeloupe, MAR Martinique, GUY French Guiana, RUN La Réunion, IDF Île-de-France, CVL Centre-Val-de-Loire, BFC Bourgogne-Franche Comté, NOR Normandie, HDF Hauts-de-France, GES Grand-Est, PDL Pays de la Loire, BRE Bretagne, NAQ Nouvelle-Aquitaine, OCC Occitanie, ARA Auvergne-Rhône-Alpes, PAC Provence-Alpes-Côtes d'Azur, COR Corse. **b** Prevalence of SARS-CoV-2 antibodies in the French population by age group and collection period. Symbols represent mean estimate and bars 95% uncertainty interval of prevalence estimate over $10^4$ iterations. **c** Prevalence of SARS-CoV-2 antibodies in the French population by region.

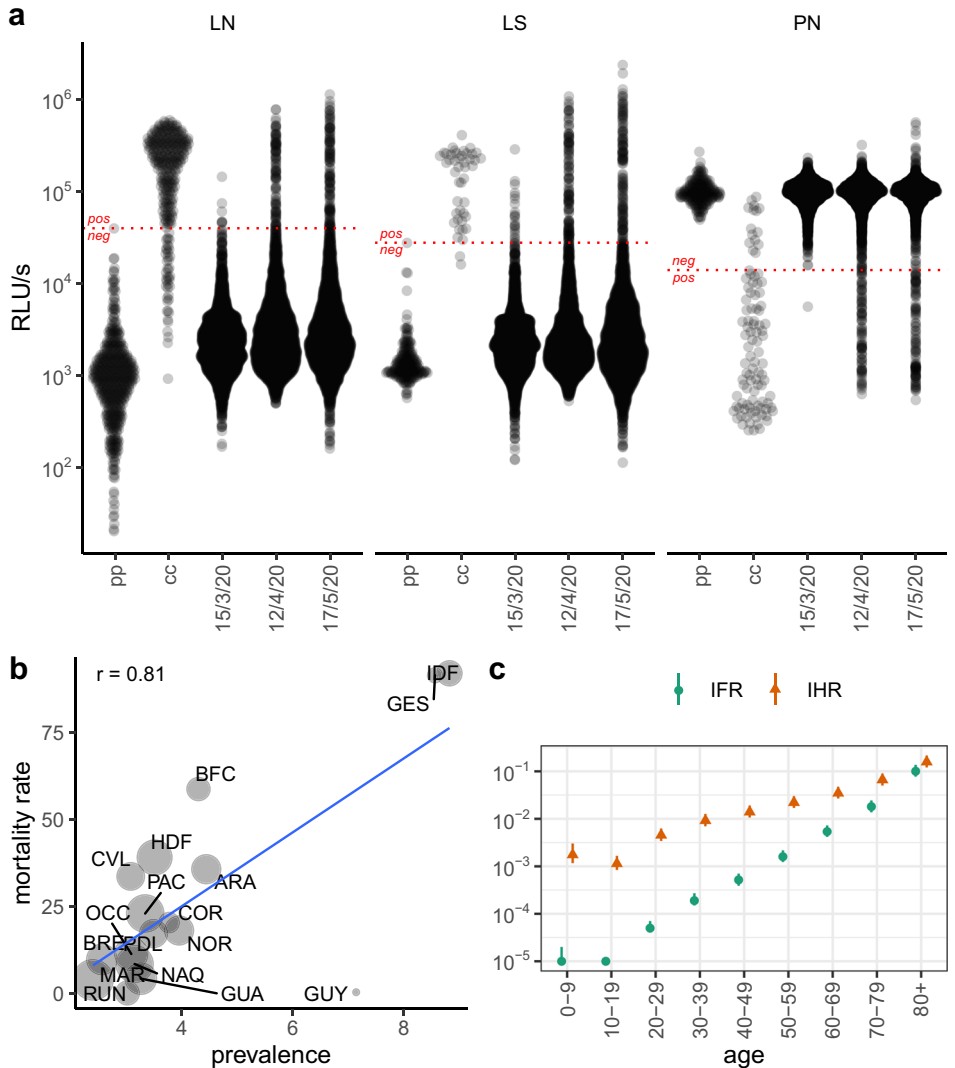

**Fig. 2 Serological assay values and population rates derived from prevalence. a** Distribution of quantitative values for the LuLISA N, LuLISA S and pseudo-neutralisation assays. Readings in relative light units (RLU in logarithmic scale) are presented for LuLISA N (LN), LuLISA S (LS) and pseudo-neutralisation (PN) assays on sera from pre-pandemic (pp) patients, confirmed cases of COVID-19 (cc), and sera sampled during three collection periods 15/3 (9–15 March 2020), 12/4 (6–12 April 2020) and 17/5 (11–17 May 2020). Positivity thresholds are indicated by horizontal dotted lines, values above the threshold indicate positivity for the LuLISA tests, whereas values below the threshold indicate positivity for the pseudo-neutralisation test. **b** Weighted correlation between estimated prevalence of SARS-CoV-2 antibodies and reported mortality rates by region. Mortality rates per 100,000 were obtained as region-specific number of deaths attributed to COVID-19 as of 29 May 2020 divided by population size. The date to account for deaths was calculated assuming that individuals with detectable antibodies at sampling time (midpoint of interval from 11 to 17 May 2020) could have been infected at minimum 15 days previously and were susceptible of dying from their infection up to 30 days post-infection. Pearson correlation coefficient (*r*) was weighted by standard error of seroprevalence estimates. Circle sizes reflect this weighting. Regions are coded as in Fig. 1a. **c** Infection fatality and infection hospitalisation rates by age. Rates are in logarithmic scale. Infection fatality rate (IFR) is estimated as the cumulative number of deaths per 100 estimated infections stratified by age. Based on available data both from French COVID-19 surveillance and published literature, we considered a lag of 20 days for both time between infection and death, and between infection and seropositivity. Infection hospitalisation rate (IHR) is calculated as the cumulative number of patients hospitalised for COVID-19 per 100 estimated infections stratified by age. We consider a time lag from infection to hospitalisation of 10 days (see "Methods").

improve their care strategies, it is essential to re-evaluate this metric as the epidemic progresses[17].

One of the primary strengths of our study is the inclusion of individuals of all ages, notably children under 10 years old. Understanding how school-aged children are susceptible to infection remains of particular importance in the face of continuing challenges for public health decisions about school settings. Seroprevalence was lowest in primary school-aged children suggesting limited susceptibility and/or transmissibility in this age group. This finding is compatible with a previous cohort study in France which

concluded that primary school-aged children were poor drivers of SARS-CoV-2 transmission amongst themselves or to teachers[18].

As expected, regional results show significantly higher seroprevalence where circulation occurred earlier and was more intense, notably in Île-de-France and Grand-Est. A large religious gathering in early March in the Grand-Est region triggered intense regional circulation of the virus and was responsible for secondary cases all over Metropolitan France and in French Guiana[19]. Estimates for other French regions confirm widespread, but less intense SARS-CoV-2 circulation at the exit of lockdown.

To date, few seroprevalence studies have included detection of neutralising antibodies, which are theoretically correlated to protection[2,3,7,10]. Importantly, seroprevalence of neutralising antibodies has not been estimated at a nationwide scale. As of 17 May 2020, we find that ~70% of seropositive individuals had detectable pseudo-neutralising antibodies with large variation across age categories and regions. Several studies similarly reported that only a fraction of seropositive individuals had detectable levels of neutralising antibodies, this fraction being variable[2,3,7–9]. This finding could be explained by differences in antibody kinetics with delayed appearance of neutralising antibodies[20].

There are three additional factors that should be taken in account in the interpretation of our results. First, we set positive thresholds for our assays to a specificity of 100%. While ruling out the risk of false positives, this could preclude the detection of the lowest antibody levels. In particular, our in-house tests were calibrated on a series of confirmed, hospitalised, COVID-19 cases, which may have limited the assessment of sensitivity. As a result, and even though the model corrected for imperfect sensitivity, we may still be underestimating the proportion of individuals with mild or asymptomatic infections who may develop a weaker or more short-lived humoral response[21–23]. Moreover, possible differential waning of antibody levels affecting mainly anti-N and pseudo-neutralising antibodies, could also result in an underestimation of seroprevalence at a distance from the epidemic waves, but this should be negligible within our relatively short surveillance period[20,24,25]. In order to facilitate the interpretation of SARS-CoV-2 infection seroprevalence as the pandemic progresses, longitudinal serological studies documenting symptoms and immune response remain essential. Finally, the urgency to provide estimates of infected population as well as logistic constraints in the lockdown period prevented the use of census or address-based sampling frames. Although the use of residual sera limits the risk of self-selection bias, it may introduce potential bias if individuals who required laboratory tests differ in terms of risk of infection from the general population. If the sampled individuals required routine monitoring for chronic health problems, they may have taken greater precautions and lowered their exposure to the virus, leading to underestimation of seroprevalence compared to the general population. However, our estimates are comparable to those reported from serological studies conducted in large preexisting representative cohorts in Île-de France, Grand-Est and Nouvelle-Aquitaine[2]. Additionally, when comparing our regional estimates and COVID-19 mortality rates, a surveillance indicator with a low susceptibility to reporting bias and which should correlate with population exposure, we find a strong correlation, with French Guiana largely influencing the overall coefficient (Fig. 2b). This discrepancy between virus circulation and mortality rate for French Guiana seems to be explained by the age structure of its infected population, skewed towards young ages[26]. These assessments against external data suggest that using residual sera can be a robust and cost-effective approach for serological surveillance.

The availability of residual sera made it possible to quickly implement sample collection early in the epidemic, providing a background seroprevalence estimate prior to the peak, and to observe epidemic dynamics throughout the first wave by including multiple collection periods. Our seroprevalence estimates, including the proportion of the population having produced pseudo-neutralising antibodies, confirmed that post-lockdown, the vast majority of the French population remained susceptible to SARS-CoV-2, even in regional hotspots. We find that a seroprevalence of at most 9% in certain regions yielded enough hospitalisations to overwhelm the healthcare system. Our results provide a critical understanding of the progression of the first epidemic wave and a framework to inform the ongoing public health response as viral transmission continues in France and globally. Serological surveillance based on residual sera will continue to be used to provide timely seroprevalence estimates as the pandemic evolves and through 2021 to monitor the progression of population level immunity and guide public health response.

## Methods

**Design and population.** Serological surveillance used repeated cross-sectional sampling of residual sera obtained from biobanks of the two largest centralising laboratories in France covering all regions and accounting for ~80% market share in specialty clinical diagnostic testing, according to the Autorité de la concurrence (French competition regulator)[27]. Residual sera included specimens from individuals of all ages undergoing routine diagnosis and monitoring in all medical specialties (such as biochemistry, immunology, allergy, etc.) except infectious diseases and obstetrics.

**Sample selection and preparation.** Specimens were collected over three 1-week periods: prior to (9–15 March 2020), during (6–12 April 2020) and following (11–17 May 2020) the nationwide lockdown. To obtain results by subgroups and enough precision, we randomly sampled available sera at the biobanks. Sampling was stratified by sex, 10-year age groups (0–9 years to ≥80 years) and region. Due to the limited number of sera available for French overseas departments (Guadeloupe, Martinique, Mayotte, French Guiana, La Réunion), all available sera were included. Relying on early modelling of the COVID-19 epidemic, which estimated an expected prevalence of 3% as of 28 March 2020, we calculated a target sample size of 3500 per collection period, with a margin of error of 0.55%[28]. After selection, blood samples were centrifuged and sera were transferred on 96-well microplates then frozen at −20 °C before transport.

**SARS-CoV-2 antibody testing.** All serological analyses were conducted with the National Reference Centre for Respiratory Infection Viruses including Influenza at the Institut Pasteur in Paris. Three novel serological assays were developed: two Luciferase-Linked ImmunoSorbent Assay (LuLISAs), detecting the nucleoprotein (LuLISA N) and spike (LuLISA S) protein of SARS-CoV-2, respectively, and a pseudo-neutralisation assay (PNT)[20,29]. The two LuLISA assays are endowed with a wide dynamic range (4-log) and a high throughput capacity (2300 assays/h)[30]. In LuLISA, the presence of all four anti-N or anti-S IgG subtypes is detected using a unique alpaca anti-human IgGVhH (single variable heavy chain antibody domain), consisting in an IgG-binding moiety directed against the Fc domain of human IgG. This VhH is expressed in fusion with the NanoKAZ luciferase, the bioluminescent activity of which is measured. The full description of the in-house anti hIgG VhH is provided in Anna et al[20]. Serum samples are considered positive when the relative light units per second (RLU/s) value is above the threshold determined for each of the LuLISA IgG/N and IgG/S assays from a pre-pandemic serum collection. The PNT mimics the SARS-CoV-2 entry step in HEK 293T cells stably expressing the human SARS-CoV-2 spike receptor ACE2 on their surface. It uses a lentiviral vector pseudo-typed with SARS-CoV-2 Spike protein, which penetrates cells in an ACE2-dependent manner, and consequently expresses a luciferase Firefly reporter. When the lentiviral Spike-mediated entry is blocked by potential serum neutralising antibodies, this leads to a reduced bioluminescent signal expressed as RLU/s. This test makes it possible to estimate the prevalence of potentially neutralising anti-S antibodies, although the effective level of protection conferred by neutralising antibodies remains unclear.

**Assay calibration.** Individual test characteristics were assessed using sets of pre-pandemic sera collected before 04/09/2019 in healthy individuals from the collection of ICAReB biobanking platform at Institut Pasteur and sera from hospitalised cases of COVID-19 confirmed by RT-PCR, with mostly moderate illness, sampled between 8 and 36 (median = 16) days past symptoms onset (Fig. 2a).

For LuLISA, serum samples are considered positive when the RLU/s value is above the threshold determined for each of the LuLISA IgG/N and IgG/S assays from a pre-pandemic serum collection. For PNT, samples are considered positive with values below a threshold set as the mean minus threefold the standard deviation determined on a collection of pre-pandemic sera assuming a normal distribution (Shapiro–Wilk normality test $W = 0.9943$, $p = ns$). This threshold permits discrimination of sera with a significant anti-SARS-CoV-2 neutralising activity from those of naïve individuals with a 99% confidence index ensuring 100% specificity on pre-pandemic sera.

In a context of low expected prevalence of infections, we set the thresholds to define a positive test result in order to obtain an in-sample empirical rate of 100% specificity to reduce the risk of false positives. This led to suboptimal sensitivities for each individual testing method, ranging from 85 to 96% (Supplementary Table 1).

Since individuals exposed to the SARS-CoV-2 virus do not undergo a single type of immune response, the results of three different but complementary

serological tests provided a more precise assessment of the population exposure to the virus. We defined seroprevalence based on the proportion of individuals who tested positive for SARS-CoV-2 antibodies for at least one of the three tests. This combination led to a perfect classification for our set of reference samples (223 pre-pandemic subjects and 45 hospitalised confirmed cases of COVID-19) (Supplementary Table 1).

**Overview of statistical methods.** Our aim is to infer the probability of SARS-CoV-2 seropositivity in the population using (1) tests results from three serological assays in specimens sampled from the population, (2) assay properties from the calibration study on known control (pre-pandemic) and case specimens and (3) post-stratification variables to account for demographics and geographic differences between the sample and population structure.

We infer seroprevalence in a Bayesian framework by fitting a general linear mixed model of seropositivity with sex, age, region and the collection period as predictors[31]. We then compute the fraction of infections reported as cases, IHR and IFR per 100 infections using national surveillance data.

**Datasets.** Our study data consist of three sets. The first contains serological results for $n$ patients along with their sex (2 levels), age class (9), region (17) and collection period (3). The second contains for the three assays, the number of pre-pandemic samples tested $N_{pp}$ of which TN have true negative results and the number of samples from confirmed cases $N_{cc}$ of which TP have true positive results. Finally, we use population counts by sex, age class and region defining 306 post-stratification cells for each collection period[32].

**Modelling seroprevalence.** First, we assume that the three serological assays performed for all specimens provide three complementary markers indicative of infection. We therefore consider a test $t$ combining the three results whereby a specimen with any positive result among the three is deemed positive, i.e. has a binary response $y_{ti} = 1$, and specimens with all three assays negative are classified by $y_{ti} = 0$. We assessed the sensitivity $se_t$ and specificity $sp_t$ of such a combined test. Let $p_t$ denote the probability of having a positive result for test $t$, test results are modelled as a Bernoulli process: $y_t \sim Bern(p_t)$.

Actual seroprevalence is derived from the frequency of positive tests, using estimates and associated uncertainties for sensitivity and specificity obtained from the calibration study. Accounting for the test performance, $p_t$ is related to the prevalence of SARS-CoV-2 antibodies $\pi$ by $p_t = se_t\pi + (1 - sp_t)(1 - \pi)$[33]. Sensitivity and specificity are defined in the following binomial processes:

$$TP_t \sim Binomial(N_{t_{cc}}, se_t), \qquad (1)$$

$$TN_t \sim Binomial(N_{t_{pp}}, sp_t), \qquad (2)$$

with subscripts pp for pre-pandemic and cc for confirmed cases.

We derived seroprevalence from regression coefficients estimated from:

$$\pi = \text{logit}^{-1}(\beta X + \alpha_{age}\sigma_{age} + \alpha_{region}\sigma_{region} + \alpha_{period}\sigma_{period}), \qquad (3)$$

where $\beta$ are fixed overall intercept and parameter for sex, with prior $\beta \sim N(0, 1)$ and $\alpha_*$ with $*$ in (age, region, period) are varying intercepts with hierarchical hyper priors:

$$\alpha_* \sim N(0, \sigma_*), \qquad (4)$$

$$\sigma_* \sim \log N(0, 1). \qquad (5)$$

We use the resulting probabilities of seropositivity in each stratum $j$ to derive poststratified estimates for the total population or by subgroups:

$$\bar{\pi} = \sum_{j=1}^{J} \frac{N_j \pi_j}{N}, \qquad (6)$$

using national census population counts $N_j$ stratified by sex, 10-year age bands and region[32].

Using posterior estimation of regression coefficients, we calculate the risk of having been infected relative to a reference category for each predictor.

The model is specified using RStan 2.21.2[34] and all data processing use R 3.6.2[35]. Code is publicly available at https://github.com/slevu/serpico2. Estimates are reported as mean of the posterior probability distributions over $10^4$ iterations and their credible intervals by the 2·5th and 97·5th percentiles.

**Fraction of reported infections.** Using seroprevalence estimates, we first infer the cumulative number of infected individuals situating their exposure 20 days prior to sampling dates. We consider a mean incubation period of 5 days[36] and a mean delay between symptoms onset (if any) and detectable seropositivity of 15 days[29]. We quantify the observable fraction of infected population from national surveillance as the ratio of documented confirmed cases reported over estimated infected individuals, accounting for a reporting delay of 10 days[37,38]. Total number of confirmed cases per day was obtained from Etalab (https://dashboard.covid19.data.gouv.fr/)[39].

**Infection fatality and infection hospitalisation rates.** We use the number of deaths stratified by age and region recorded in hospitals[40] and overall deaths in nursing homes (obtained from national surveillance[37]) to derive the IFR by age. Age distribution of deaths in nursing homes during the first epidemic wave was obtained separately from a sample of 312 facilities. Dates of death events were considered with a time lag from infection to death of 20 days[14,38]. Hospital admission data were obtained from national surveillance[37] considering a delay from infection to hospitalisation of 10 days[38].

**Ethical considerations.** Authorisation for conservation and preparation of elements of the human body for scientific use was granted to the two biobanks by the bioethics committee from General Board for Research and Innovation (DGRI) of French Ministry of Higher Education and Research (approvals Nos. AC-2015-2418 and AC-2018-3329). Information regarding secondary use of de-identified residual sera for approved research studies was systematically displayed and orally communicated at the primary clinical laboratories. The Ethics Committee (Comité de Protection des Personnes Ile-de-France VI, CHU Pitié-Salpétrière Hospital, Paris, France) waived the need for ethical approval for the collection, analysis and publication of the retrospectively obtained and anonymized specimens and data for this study. This work was carried out following regulations of the French Public Health Code (articles L. 1413-7 and L. 1413-8) and the French Commission for Data Protection (CNIL).

**Reporting summary.** Further information on experimental design is available in the Nature Research Reporting Summary linked to this paper.

## Data availability
All data are present in the article and its Supplementary Information files or upon reasonable request from the corresponding author, although requests for data might require partial aggregation or downsampling to protect patient privacy. Source data are provided with this paper.

## Code availability
The Zenodo repository https://doi.org/10.5281/zenodo.4586147 includes code to reproduce analyses presented in the paper[41].

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

## Acknowledgements
We thank Christine Larsen, Bruno Coignard and Jean-Claude Desenclos (Santé publique France) for their helpful contribution at setting up the study; from Institut Pasteur, Isabelle Cailleau for support in the funding process, Hélène Munier-Lehmann for access to automate and supply management at the Unit of Chemistry and Biocatalysis, Yves L. Janin for providing the luciferase prosubstrate hikarazine 108, Philippe Souque for production of the lentiviral pseudo-types, the whole ICAReB team and COVID-19 support staff for sample management at Institut Pasteur; Juliette Paireau (Institut Pasteur, Santé publique France), Rodolphe Thiébaut (Bordeaux Université) and Xavier de Lamballerie (Aix-Marseille Université) for valuable comments on first results. We also thank the team from the Eurofins Biomnis Sample Library and from CerbaHealthcare Benedicte Roquebert (Laboratoire Cerba) and Marie Pierre Guerra (CerbaXpert) for contributing to sample collection. Santé publique France provided funding to the NRC and to the two centralising biobanks to cover sample collection, preparation, transport and analysis costs. The funder had no role in analysis, interpretation of data or writing of the report. S.L.V., G.J. and H.N. had full access to all the data and had responsibility to submit for publication.

## Author contributions
H.N., G.J., J.-B.R, S.B.-S., D.L.-B. and S.v.d.W. conceived and planned the study. V.M., C.R., M.-N.U., L.P.d.F. and O.H. contributed to sample collection. T.R., F.A., S.G., P.C., S.P., N.E. and M.G. developed the assays and carried out the biological analyses. S.L.V., J.-B.R., H.N. and G.J. designed and performed data analyses. H.N., G.J., J.-B.R., S.L.V., S.B.-S., D.L.-B., T.R., P.C., F.A., C.D., S.v.d.W., L.L., Y.G. and L.F. contributed to the interpretation of results. H.N., G.J. and S.L.V. wrote the draft manuscript. All authors discussed the results and approved the final manuscript.

## Competing interests
S.G. and T.R. declare patents for the proluciferins (hikarazines) synthesis and uses ("Imidazopyrazine derivatives, process for preparation thereof and their uses as luciferins", EP 3395803/WO 2018197727, 2018), and have applied for a patent that includes claims describing the LuLISA (EP20315224.4) used in this study. F.A. and P.C. have applied for a patent claiming the PNT ("High throughput methods and products for SARS-CoV-2 sero-neutralization assay" US 63/107,896). N.E., M.G., C.D. and S.v.d.W. declare patents pending for the SARS-associated coronavirus diagnostics ("Severe acute respiratory syndrome (SARS) associated coronavirus diagnostics", US 10,948,490 B1) related to this study. S.v.d.W. and N.E. declare patents issued ("Use of proteins and peptides coded by the genome of a novel strain of SARS-associated coronavirus" EP 1697507; and "Novel strain of SARS-associated coronavirus and applications thereof" EP1694829) not directly related to this study. S.v.d.W. is a board member (non-financial support) for the International Society for Influenza and other Respiratory Virus Diseases and has a patent application filed ("Methods and reagents for the specific and sensitive detection of SARS-CoV-2", PCT/EP2020055939 and US16/809,717), not directly related to this study. The other authors declare no competing interests.
