## [Peer Review File · Nature Communications]

REVIEWERS' COMMENTS

Reviewer #1 (Remarks to the Author):

The authors have addressed most of my comments.

Minor comments

1. Table S2, please add the corresponding statistics for France for comparison, such as age, sex, and regions. This will help understand the study representativeness.

Reviewer #2 (Remarks to the Author):

The authors did a nice job responding to previous comments and I have no further concerns with the manuscript above and beyond the limitations of the potential non-representativeness of the sample and small assay positive control set based on hospitalised cases followed for very little time post infection. These limitations are adequately discussed in the paper.

Reviewer #3 (Remarks to the Author):

Author:

We would refer to the first response to reviewer 2 for a similar comment. Use of residual sera allowed our study to cover all age groups (including young children), the entire French population including overseas departments, and with rapid implementation of sample collection in order to provide seroprevalence estimates early in the first wave. Other approaches, such as blood donors or sampling based on voluntary screening, would not fulfill all these objectives.

Reviewer reply:

Coverage of all age groups is not sufficient to ensure that seroprevalence estimates are not biased, even after adjusting for (e.g.) non-response. I agree that sampling on (e.g.) voluntary screening would not address the issue, but other sampling approaches could. For example, address-based sampling could have avoided the potential biases induced by sampling within biobanks. Such survey studies are difficult and costly to implement, but they are potentially superior to the use of residual sera.

Author:

The threshold was statistically determined using raw values for a set of pre-pandemic sera. The threshold value is calculated on the mean raw values for the whole set of pre-pandemic sera minus 3 Standard deviations corresponding to a confidence index of 99%. We hypothesize that people with a luminescence value below the threshold statistically have a sufficiently different neutralizing response from pre-pandemic individuals to conclude that they have generated neutralizing antibodies following SARS-CoV-2 infection.

On pre-pandemic sera this threshold resulted in a 100% specificity. A higher confidence index was applied for the pseudo-neutralization assay to prevent a bias due to the smaller set of sera from positive COVID-19 cases used for the pseudo-neutralization tests in comparison to the LuLISA assay.

This was further specified in the manuscript in the Methods, SARS-CoV-2 antibody testing section as follows: "When the lentiviral Spike-mediated entry is blocked by potential serum neutralising antibodies, this leads to a reduced bioluminescence signal expressed as RLU/s and samples are

considered positive with values below a threshold set as the mean minus 3- fold the standard deviation determined on a collection of pre-pandemic sera. This threshold ensures 100% specificity with a high confidence index to prevent any bias due to the smaller proportion of sera positive in pseudo-neutralization in the population studied in comparison to the proportion of seroconversion detected by the LuLISA assays."

Reviewer reply:

In the assay calibration section, I would suggest a slight rephrasing of the sentence "in order to obtain a 100% specificity to reduce the risk of false positives" into "in order to obtain an in-sample empirical rate of 100% specificity to reduce the risk of false positives".

The authors responded that the threshold was defined as "mean - 3*std dev corresponding to a confidence index of 99%". Firstly, the 99% confidence is only valid under certain assumptions (e.g., normality). It's unclear that this assumption holds here, but that's a minor point. Second, it does not look like this rule is explained anywhere in the manuscript, unless I missed it. It should be included in the "assay calibration" section. Third, when looking at figure S1, it seems like the thresholds for the first two assays (LS and LN) were defined as the largest values in pre-pandemic samples. This should be clarified.

In summary, whether the study provides "good" estimates of seroprevalence can be debated, but I am willing to believe that the estimates are reasonably close to the (true) seroprevalences at the beginning of the pandemics. I would say that the authors just need to be a bit more transparent about the process they used in their data analysis (here, i am referring to the definition of thresholds).

REVIEWERS' COMMENTS

Reviewer #1 (Remarks to the Author):

The authors have addressed most of my comments.

Minor comments

1. Table S2, please add the corresponding statistics for France for comparison, such as age, sex, and regions. This will help understand the study representativeness.

Author reply:

We thank Reviewer#1 for this suggestion. Table S2 has been modified accordingly.

Reviewer #2 (Remarks to the Author):

The authors did a nice job responding to previous comments and I have no further concerns with the manuscript above and beyond the limitations of the potential non-representativeness of the sample and small assay positive control set based on hospitalised cases followed for very little time post infection. These limitations are adequately discussed in the paper.

Author reply:

We are delighted that our discussion on the limitations regarding the representativeness of the sample and the assay positive control group adequately addressed the reservations of Reviewer #2.

Reviewer #3 (Remarks to the Author):

Author:

We would refer to the first response to reviewer 2 for a similar comment. Use of residual sera allowed our study to cover all age groups (including young children), the entire French population including overseas departments, and with rapid implementation of sample collection in order to provide seroprevalence estimates early in the first wave. Other approaches, such as blood donors or sampling based on voluntary screening, would not fulfill all these objectives.

Reviewer reply:

Coverage of all age groups is not sufficient to ensure that seroprevalence estimates are not biased, even after adjusting for (e.g.) non-response. I agree that sampling on (e.g.) voluntary screening would not address the issue, but other sampling approaches could. For example, address-based sampling could have avoided the potential biases induced by sampling within biobanks. Such survey studies are difficult and costly to implement, but they are potentially superior to the use of residual sera.

Author reply:

We agree with Reviewer #3 regarding the potential superiority of other sampling methods. However, these sampling strategies were not compatible with our study objectives, in particular a collection of population specimens early in the epidemic and prior to first national lockdown. To our knowledge, no address-based or telephone-based sampling survey have been put in place to inform the serostatus of a country population as early as February

2020. In France at least, we experienced a major disruption in the organizational capacity for research projects, notably survey researchers had to adapt to remote working which impeded the rapid implementation of some superior survey methods.

We added a sentence in the discussion to further stress this limitation :

“Finally, the urgency to provide estimates of infected population as well as logistic constraints in the lockdown period prevented the use of census or address-based sampling frames.”

Author:

The threshold was statistically determined using raw values for a set of pre-pandemic sera. The threshold value is calculated on the mean raw values for the whole set of pre-pandemic sera minus 3 Standard deviations corresponding to a confidence index of 99%. We hypothesize that people with a luminescence value below the threshold statistically have a sufficiently different neutralizing response from pre-pandemic individuals to conclude that they have generated neutralizing antibodies following SARS-CoV-2 infection.

On pre-pandemic sera this threshold resulted in a 100% specificity. A higher confidence index was applied for the pseudo-neutralization assay to prevent a bias due to the smaller set of sera from positive COVID-19 cases used for the pseudo-neutralization tests in comparison to the LuLISA assay.

This was further specified in the manuscript in the Methods, SARS-CoV-2 antibody testing section as follows: “When the lentiviral Spike-mediated entry is blocked by potential serum neutralising antibodies, this leads to a reduced bioluminescence signal expressed as RLU/s and samples are considered positive with values below a threshold set as the mean minus 3-fold the standard deviation determined on a collection of pre-pandemic sera. This threshold ensures 100% specificity with a high confidence index to prevent any bias due to the smaller proportion of sera positive in pseudo-neutralization in the population studied in comparison to the proportion of seroconversion detected by the LuLISA assays.”

Reviewer reply:

In the assay calibration section, I would suggest a slight rephrasing of the sentence “in order to obtain a 100% specificity to reduce the risk of false positives” into “in order to obtain an in-sample empirical rate of 100% specificity to reduce the risk of false positives”.

Author reply:

We are in agreement with the reviewer’s suggestion and have modified the sentence as indicated.

The authors responded that the threshold was defined as “mean – 3*std dev corresponding to a confidence index of 99%”. Firstly, the 99% confidence is only valid under certain assumptions (e.g., normality). It’s unclear that this assumption holds here, but that’s a minor point.

Author reply:

We thank the reviewer for pointing this out. The normal distribution was assessed on pre-pandemic sera using the Shapiro-Wilk test. The text has been modified to include this information.

Second, it does not look like this rule is explained anywhere in the manuscript, unless I missed it. It should be included in the "assay calibration" section.

Third, when looking at figure S1, it seems like the thresholds for the first two assays (LS and LN) were defined as the largest values in pre-pandemic samples. This should be clarified.

Author reply:

For clarification the text of the material and methods section was slightly reorganized and additional information was added under the "assay calibration" section.

In summary, whether the study provides "good" estimates of seroprevalence can be debated, but I am willing to believe that the estimates are reasonably close to the (true) seroprevalences at the beginning of the pandemics. I would say that the authors just need to be a bit more transparent about the process they used in their data analysis (here, i am referring to the definition of thresholds).

Author reply:

We would like to thank Reviewer #3 for a thorough review with helpful comments that improved the presentation of our study.